# Metformin and the Development of Asthma in Patients with Type 2 Diabetes

**DOI:** 10.3390/ijerph19138211

**Published:** 2022-07-05

**Authors:** Fu-Shun Yen, Chih-Cheng Hsu, Ying-Hsiu Shih, Wei-Lin Pan, James Cheng-Chung Wei, Chii-Min Hwu

**Affiliations:** 1Dr. Yen’s Clinic, Taoyuan 33354, Taiwan; yenfushun@gmail.com; 2Institute of Population Health Sciences, National Health Research Institutes, Miaoli County 35053, Taiwan; cch@nhri.edu.tw; 3Department of Health Services Administration, China Medical University, Taichung 40402, Taiwan; 4Department of Family Medicine, Min-Sheng General Hospital, Taoyuan 33044, Taiwan; 5Management Office for Health Data, China Medical University Hospital, Taichung 40459, Taiwan; hsiu.cmuh@gmail.com; 6College of Medicine, China Medical University, Taichung 40201, Taiwan; 7Department of Internal Medicine, Mackay Memorial Hospital, Taipei 10449, Taiwan; kpsmile01058@gmail.com; 8Department of Allergy, Immunology & Rheumatology, Chung Shan Medical University Hospital, Taichung 40201, Taiwan; 9Institute of Medicine, Chung Shan Medical University, Taichung 40201, Taiwan; 10Graduate Institute of Integrated Medicine, China Medical University, Taichung 40402, Taiwan; 11Faculty of Medicine, National Yang-Ming University School of Medicine, Taipei 11221, Taiwan; 12Section of Endocrinology and Metabolism, Department of Medicine, Taipei Veterans General Hospital, Taipei 11217, Taiwan

**Keywords:** type 2 diabetes, metformin, asthma, exacerbation, hospitalization

## Abstract

We conducted this study to compare the risks of asthma development and exacerbation between metformin users and nonusers. Overall, 57,743 propensity score-matched metformin users and nonusers were identified from Taiwan’s National Health Insurance Research Database between 1 January 2000, and 31 December 2017. We used the Cox proportional hazards model with robust standard error estimates to compare the risks of asthma onset, exacerbation, and hospitalization for asthma in participants with type 2 diabetes (T2D). Compared with metformin nonuse, the aHRs (95% CI) for metformin use in asthma development, exacerbation, and hospitalization for asthma were 1.13 (1.06–1.2), 1.62 (1.35–1.95), and 1.5 (1.22–1.85), respectively. The cumulative incidences of asthma development, exacerbation, and hospitalization for asthma were significantly higher in metformin users than nonusers (*p* < 0.001). A longer cumulative duration of metformin use for more than 728 days was associated with significantly higher risks of outcomes than metformin nonuse. Our study demonstrated that metformin users showed significantly higher risks of asthma development, exacerbation, and hospitalization for asthma than metformin nonusers. Moreover, metformin use for more than 728 days was associated with higher risks of outcomes. A randomized control study is warranted to verify our results.

## 1. Introduction

The function of the lung is to transport oxygen from inhaled air into the bloodstream and remove carbon dioxide from circulation [1]. However, some genetic or environmental factors may produce airway hyperresponsiveness, chronic inflammation, and remodeling, causing asthma with increased airway resistance, limitation, and variable respiratory symptoms (characterized by brief and dramatic attacks) [2]. Asthma is one of the most common non-communicable chronic diseases worldwide [2]. There are approximately 262 million patients with asthma worldwide presently, with a prevalence rate of about 3.53% [3]. Taiwan has about 737,000 patients with asthma, with a prevalence rate of about 3.26% [3]. Obesity is an important risk factor for asthma [4]. Patients with diabetes are often obese, and the metabolic abnormalities associated with diabetes and truncal adiposity may impact pulmonary physiology and increase asthma risk [5,6]. As diabetes can affect respiratory function and increase the risk of infection, it may aggravate asthma severity and exacerbations [7].

Metformin is the most commonly used antidiabetic drug worldwide. It can reduce blood sugar and modulate metabolism by inhibiting mitochondrial respiratory-chain complex-1 and activating the adenosine monophosphate (AMP)-activated protein kinase (AMPK) [8]. It can suppress IgE-mediated mast cell activation [9] and increase the ratio of lymphocyte T regulatory (Treg)/lymphocyte T helper 17 (Th17) cells in mice [8]. Metformin can downregulate the levels of pro-inflammatory cytokines, such as tumor necrosis factor (TNF)-α, interleukin (IL)-1β, IL-4, and IL-6 [8]. It can decrease eosinophilic airway inflammation in mice lung tissue and reduce smooth muscle proliferation in cultured murine airway smooth muscle cells [10,11]. These preclinical data suggest that metformin may prevent the development of asthma.

Two clinical studies have shown that metformin use is associated with a lower risk of asthma onset [12,13]. Two retrospective cohort studies have also shown that metformin may decrease the risk of asthma exacerbation [14,15]. However, the role of metformin in asthma has been inconclusive due to few studies and methodological bias in research [8]. Therefore, we conducted this study to investigate the impact of metformin on the risks of asthma development and exacerbation using a more stringent methodology in patients with type 2 diabetes.

## 2. Materials and Methods

### 2.1. Study Population and Data Source

Taiwan implemented the National Health Insurance (NHI) program in 1995. The government established the Bureau of National Health Insurance to execute this plan. It is a compulsory insurance system; the government is a single buyer, and the people only need to pay a small premium and copayment. By 2000, about 99% of all Taiwanese people participated in the NHI program [16]. Information of insured persons, including age, sex, area of residence, insured premium, diagnosis, medications, examinations, and medical procedures, is recorded in the NHI Research Database (NHIRD). The diagnosis is based on the International Classification of Diseases, Ninth and Tenth Revision, Clinical Modification (ICD-9-CM and ICD-10-CM), and the NHIRD is linked to the National Death Registry to provide death information. The study was conducted in accordance with the Declaration of Helsinki, and was approved by the Research Ethics Committee of China Medical University and Hospital (CMUH109-109-REC2-031). Identifiable information of participants and care providers was encrypted and scrambled before release to protect individual privacy. Informed consent was waived by the Research Ethics Committee.

### 2.2. Study Design and Participants

Participants with newly diagnosed type 2 diabetes (T2D) were identified between 1 January 2000, and 31 December 2017, and followed up to 31 December 2018. The diagnosis of T2D was based on the ICD codes (ICD-9 code: 250, except 250.1×; ICD-10: E11. Appendix A) for at least 3 outpatient visits or one hospitalization record. The algorithm of using ICD codes to classify T2D was validated by a Taiwanese study with an accuracy of 74.6% [17]. Exclusion criteria were as follows (Figure 1): (1) missing data on age or gender; (2) age below 20 or above 80 years; (3) diagnosis of type 1 diabetes, asthma, hepatic failure, or received dialysis; and (4) previous diagnosis of T2D before 1 January 2000, to exclude prevalent T2D cases.

### 2.3. Procedures

We defined the first date of metformin use after the diagnosis of T2D as the index date. Participants who received metformin treatment were study cases and those who never received metformin during the follow-up time served as controls. The index date of the control participant was randomly assigned as the same period of T2D diagnosis to metformin use of the paired metformin user. Some relevant variates were observed and matched between metformin users and nonusers, including sex, age, obesity, smoking status, alcohol-related disorders, hypertension (HT), dyslipidemia, coronary artery disease (CAD), stroke, atrial fibrillation, peripheral arterial occlusive disease (PAOD), chronic kidney disease (CKD), rheumatoid arthritis, systemic lupus erythematosus, liver cirrhosis, cancers, psychosis, depression, dementia, chronic obstructive pulmonary disease (COPD), and heart failure, diagnosed within 1 year before the index date. Medications used during the follow-up period, such as oral antidiabetic drugs, insulin, immunosuppressants, statin, and aspirin, were also included. We used the Charlson Comorbidity Index (CCI), Diabetes Complication Severity Index (DCSI) score [18,19], and the number of oral antidiabetic drugs to evaluate T2D severity.

### 2.4. Main Outcomes

We observed and compared the incidence rates of newly diagnosed asthma, acute asthma exacerbation, and hospitalization for asthma during the follow-up period. We identified cases of asthma using diagnostic codes with at least three outpatient visits or one hospitalization [20]. Asthma exacerbation was defined when a patient received systemic corticosteroids or had an asthma-related emergency room visit or hospital admission during the follow-up period [21].

### 2.5. Statistical Analysis

Propensity-score matching was used to optimize the relevant variables between metformin users and nonusers [22]. We estimated the propensity score for each participant using non-parsimonious multivariable logistic regression with metformin use as the dependent variable, and the group of metformin no-use was matched without replacement. We included 42 clinically related covariates, such as sex, age, obesity, smoking, comorbidities, CCI and DCSI scores, the number and item of oral anti-diabetic drugs, insulin, statin, aspirin, immunosuppressants, influenza vaccination, adult health examination, HbA1C > 2 times per year, as independent variables (Table 1). The nearest-neighbor algorithm was used to match pairs, standardized mean difference (SMD) was used to assess their discrimination, and assuming the SMD value < 0.1 to be a negligible difference between the study and control cohorts.

We calculated the incidence rate of asthma per 1000 person-years of follow-up for the study and control groups. Crude and multivariable-adjusted Cox proportional hazards models were used to compare the outcomes between the two groups. The results are expressed as hazard ratios (HRs) and 95% confidence intervals (CIs) for metformin users compared with nonusers. To assess the investigated risks, we censored participants on the date of respective outcomes, death, or at the end of the follow-up on 31 December 2018, whichever came first. The Kaplan–Meier method and log-rank tests were used to compare the cumulative incidence of asthma development, exacerbation, and hospitalization for asthma between metformin users and nonusers during the follow-up period. We also assessed the cumulative duration of metformin use for the risks of asthma development, exacerbation, and hospitalization for asthma compared with metformin nonuse.

SAS (version 9.4; SAS Institute, Cary, NC, USA) was used for statistical analysis, and a two-tailed *p* value < 0.05 was considered significant.

## 3. Results

### 3.1. Participants

From 1 January 2000, to 31 December 2017, we identified 258,339 participants with newly diagnosed T2D. Of these, 167,890 were metformin users, and 90,449 were nonusers (Figure 1). After excluding ineligible participants, 1:1 propensity score matching was used to construct 57,743 pairs of metformin users and nonusers. In the matched cohorts (Table 1), 50.47% of participants were female; the mean (SD) age was 57.0 (12.89) years. The mean follow-up time for metformin users and nonusers was 6.67 (4.41) and 4.15 (3.89) years, respectively.

### 3.2. Main Outcomes

In the matched cohorts (Table 2), 2505 (4.34%) metformin users and 1440 (2.49%) nonusers developed asthma during the follow-up period (incidence rate: 6.66 vs. 6.11 per 1000 person-years). In the multivariable model, metformin users showed a 13% higher risk of asthma development (aHR = 1.13, 95% CI = 1.06–1.2) than nonusers (Table 2). Participants with older age, coronary artery disease, atrial fibrillation, rheumatoid arthritis, depression, COPD, heart failure, and sulfonylurea use showed a higher risk of incident asthma (Table 2), and men, persons with dyslipidemia, CCI score > 3, and statin use showed a lower risk of incident asthma (Table 2).

The multivariable model also exhibited that metformin users showed a 62% higher risk of asthma exacerbation (aHR = 1.62, 95% CI = 1.35–1.95) and a 50% higher risk of hospitalization for asthma (aHR = 1.5, 95% CI = 1.22–1.85) than nonusers (Table 3).

The Kaplan–Meier analysis showed that the cumulative incidences of asthma development, acute exacerbation of asthma, and hospitalization for asthma were significantly higher in metformin users than nonusers (Log-rank test *p*-value < 0.001; Figure 2).

### 3.3. Cumulative Duration of Metformin Use

We observed the association between the cumulative duration of metformin use and the risk of asthma onset, exacerbation, and hospitalization for asthma (Table 4). Metformin use for more than 728 days was associated with significantly higher risks of asthma onset, exacerbation, and hospitalization for asthma than metformin nonuse (Table 4).

## 4. Discussion

This study demonstrated that metformin use was associated with significantly higher risks of asthma development, exacerbation, and hospitalization for asthma compared with metformin nonuse in patients with type 2 diabetes. Moreover, metformin use for more than 728 days (nearly two years) was associated with significantly higher risks of incident asthma, acute exacerbation, and hospitalization for asthma than metformin nonuse.

Chen et al. conducted a retrospective cohort Taiwan-based study to compare the risk of incident asthma between persons with and without T2D. Results showed that persons with T2D had a 30% higher risk of asthma (aHR 1.30, 95% CI 1.24–1.38). They conducted a nested case–control study from the above cohort to investigate the risk of incident asthma with different antidiabetic drugs. Results showed that insulin was associated with a higher risk of asthma (adjusted odds ratio (aOR) 2.23 95% CI 1.52–3.58), and metformin was associated with a 25% lower risk of asthma (aOR 0.75, 95% CI 0.60–0.95) [12]. Rayner et al. conducted a retrospective cohort study with propensity score matching using the United Kingdom primary care database to compare the risk of incident asthma between antidiabetic drugs and found that insulin (aHR 1.25, 95% CI 1.01–1.56) was associated with a higher risk, and metformin (aHR 0.80, 95% CI 0.69–0.93) and sulfonylureas (aHR 0.76, 95% CI 0.60–0.97) were associated with a lower risk of incident asthma [13]. We conducted a retrospective cohort study with propensity score matching using Taiwan’s NHIRD and showed that metformin use was associated with a higher risk of asthma development than metformin nonuse (aHR 1.13 95% CI 1.06–1.2). The possible reasons for the differences between these three studies are as follows: (1) the study population is different (The study by Chen and the present study are both Taiwan-based; the study by Rayner is UK-based). (2) The study methods are dissimilar (Chen conducted a nested case–control study; Rayner and the present study used a propensity score-matched cohort). (3) The matched population is different. Chen and Rayner matched diabetic persons with non-diabetic persons to assess the impact of diabetes on the risk of asthma and then used multivariable analysis to observe the secondary outcomes of different antidiabetic drugs impacting asthma incidence. We matched metformin users with nonusers in participants with T2D to assess the primary outcome of asthma development. We have performed well matching of critical variables including age, sex, obesity, smoking, comorbidities and medications to decrease the measurable bias. Persons who received metformin may have health-seeking behaviors, such as regular clinic visits, check-ups, and adherence to medications, and this healthy user effect may influence observed outcomes between metformin users and nonusers [23]. We matched the use of influenza vaccination, adult health examination, and the number of HbA1C tests per year to reduce the bias of the healthy user effect [24]. Additionally, this study showed that the longer cumulative duration of metformin use was associated with higher risk of asthma development.

Preclinical studies have demonstrated that metformin can stabilize mast cells, upregulate the ratios of Th1/Th2 and Treg/Th17 cells, downregulate inflammatory cytokines (IL-4, IL-5, IL-6, IL-13, IL-17), and increase anti-inflammatory cytokines (IL-10) with the possibility of attenuating asthma development [8,9]. However, some studies have shown conflicting results. Saenwongsa et al. evaluated interferon (IFN)-α expression after antidiabetic treatment in patients with T2D. They found suppressed IFN-α expression, via decreased mechanistic target of rapamycin (mTOR)1 signaling, in patients with T2D using metformin, which probably reduced selection intensity of germinal centers and affinity maturation of immune cells [25]. Shore et al. investigated the role of metformin in altering the pulmonary phenotype of *db*/*db* mice after ozone (O3) exposure and demonstrated that metformin treatment could not alter innate airway hyperresponsivess and increased pulmonary inflammatory responses to ozone [26]. More research is needed on this subject.

This study revealed that older participants, females, comorbidities, such as coronary artery disease, heart failure, atrial fibrillation, chronic kidney disease, COPD, and rheumatoid arthritis, showed a significantly higher risk of asthma onset. Statin use was associated with lower asthma risk. Previous studies have shown that older adults, females, COPD, and rheumatoid arthritis showed a higher risk, and statin use had a lower risk of incident asthma [24,27,28,29]. Reports suggest that asthma, cardiovascular disease, and chronic kidney disease are closely related, probably due to the contribution of chronic systemic inflammation [30]. However, most studies reported that asthma could cause cardiovascular disease or chronic kidney disease [31]. Few studies have shown that cardiovascular disease or chronic kidney disease may increase the incidence of asthma. Future studies on this topic are needed.

Li et al. conducted an excellent cohort study with propensity score matching and a new user design to evaluate metformin use and asthma outcomes in Taiwanese subjects with concurrent asthma and diabetes [14]. They reported that metformin users had a lower risk of asthma-related hospitalization (aOR = 0.21, 95% CI:0.07–0.63) and exacerbation (aOR = 0.39, 95% CI: 0.19–0.79) than nonusers. Wu et al. used the IBM MarketScan Research database of the United States to conduct a retrospective cohort study and demonstrated that metformin initiation was associated with lower risks of asthma exacerbation (aHR 0.92, 95% CI 0.86–0.98) and hospitalization (aHR 0.67, 95% CI, 0.50–0.91) [15]. However, in our propensity score-matched cohort study, metformin use was associated with a higher risk of asthma exacerbation and hospitalization. The conflicting results could be explained as follows: (1) difference in the number of participants (Li, 444; Wu, 14,641; present study, 57,743 metformin users); (2) differences in matching variables; (3) racial differences among participants (Li and the present study: Chinese subjects; Wu: American subjects). Metformin is associated with a higher mortality risk in patients with end-stage renal disease [32] and a higher risk of lactic acidosis in patients with advanced heart failure or liver cirrhosis [33]. Our previous study of metformin use in patients with COPD has disclosed that metformin was associated with a higher risk of hospitalization for COPD and invasive mechanical ventilation [34]. Physicians may choose to switch from metformin to other antidiabetic drugs in patients with advanced renal, liver, heart diseases, or frailty [24]. We excluded patients receiving dialysis or those with hepatic failure and matched patients with chronic kidney diseases, alcohol-related disorders, liver cirrhosis, and chronic obstructive pulmonary disease to decrease the bias of confounding by indication. A prevalent user may be more tolerant or adherent to the medication under study than a new user and influence the investigated results. We used a new user design in this study to mitigate the bias of the healthy user effect [24]. Metformin can inhibit the complex I of the mitochondrial electron transport chain and reduce ATP production, leading to energy stress and diminished mitochondrial respiration in the skeletal muscle [35]. Long-term metformin use may lower serum vitamin B12 levels [36] and influence respiratory muscle function [36]. Both these effects from metformin use may increase the risks of asthma exacerbation and hospitalization.

The strength of this study is that it is a large population-based study with 57,743 paired participants. It covered an 18-year follow-up period from 1 January 2000, to 31 December 2018, and included critical variables (demographic, comorbidities, medications, procedures, and vaccination), which may influence outcomes. This study also has several limitations. First, this dataset lacked lifestyle, education, family income, marital status, body mass index, smoking status, alcohol drinking, occupational exposure, and family history details. It also lacked data on air pollution, blood cultures, biochemical tests, hemoglobin A1C, allergens, and pulmonary function tests, precluding an accurate diagnosis and evaluation of asthma and T2D. However, we matched many crucial variables, such as age, sex, the ICD codings of smoking, alcohol-related disorders, obesity, influenza vaccination, adult health examination, HbA1C test, comorbidities, CCI, and medications, for a maximal balance between study and comparison groups; we also matched the item and number of oral antidiabetic drugs, insulin, and DCSI scores to balance diabetes severity and increase comparability. Second, as our database does not have access to different immune cells from the sputum or blood, we cannot distinguish the various endotypes of asthma. Third, we do not know whether patients are taking medications as prescribed by their doctors from this dataset. There is also no information of patients’ preferences or concerns about various antidiabetic drugs, and physicians’ preferences of prescription. Though metformin is the recommended first-line treatment for T2D, it is only emphasized in recent several years. As our study was conducted from 2000 to 2018, many physicians may prefer to use sulfonylurea to treat T2D in 2000 to 2008 for its fast glucose lowering effect. These biases may affect the results of this study. Fourth, this study was conducted on Taiwanese subjects, and hence, the results may not apply to other races. Finally, a cohort study is usually subject to some unmeasured and unknown confounding factors, and a randomized control trial is needed to verify our results.

## 5. Conclusions

Persons with T2D are often obese with metabolic syndrome, and hence, they may be prone to asthma development. Diabetes can also aggravate asthma severity and exacerbation. Several studies suggest that metformin may reduce the occurrence and exacerbation of asthma. However, our research (after adjusting the patient’s health behavior) showed that metformin was associated with higher risks of asthma development and exacerbation. Randomized control studies are warranted to resolve this concern.

## Figures and Tables

**Figure 1 ijerph-19-08211-f001:**
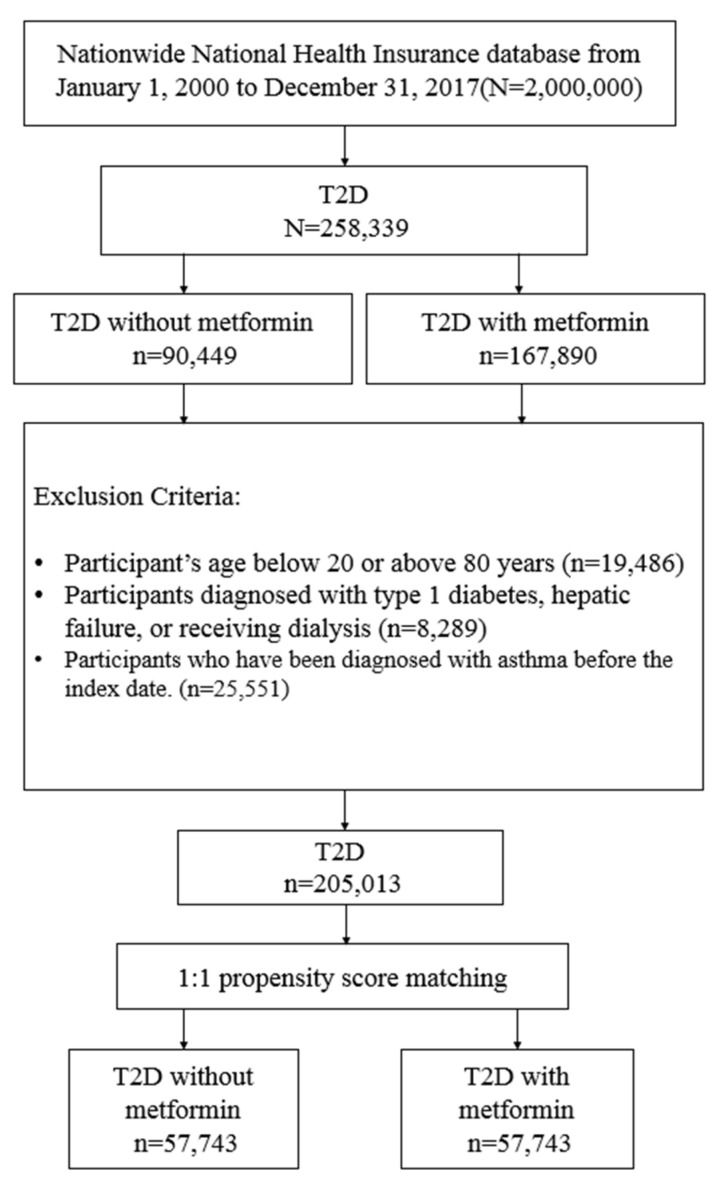
Process flow chart for the study.

**Figure 2 ijerph-19-08211-f002:**
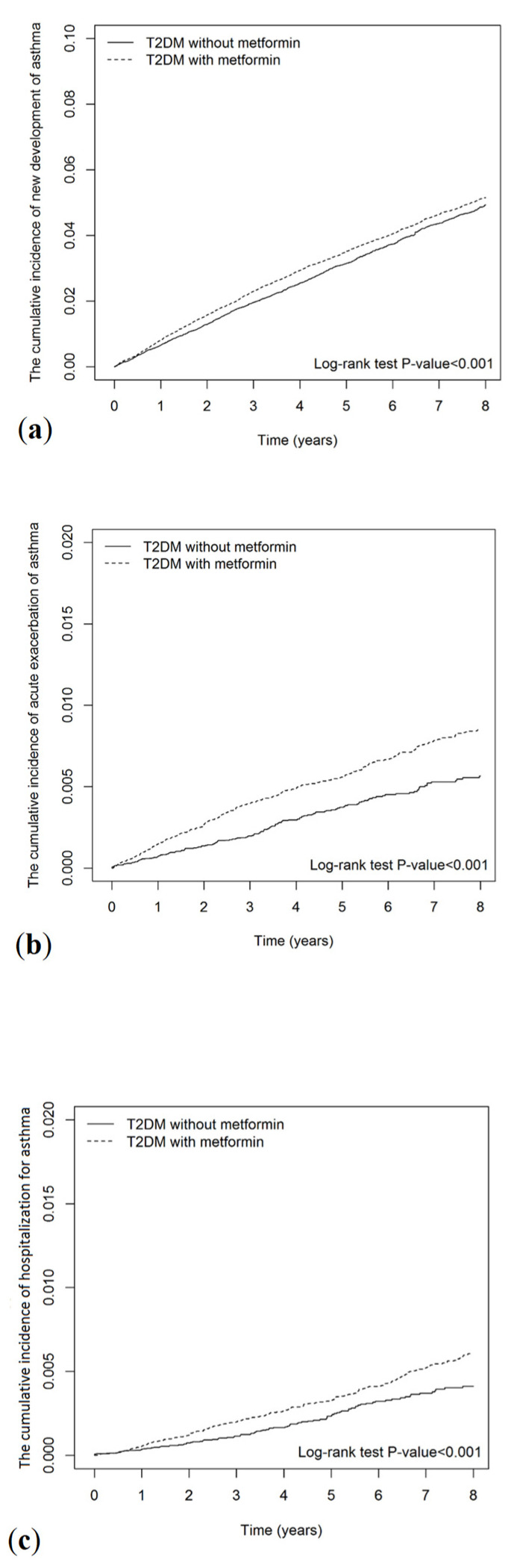
Cumulative incidences of (**a**) new development of asthma, (**b**) acute exacerbation of asthma, and (**c**) hospitalization for asthma for metformin users vs. nonusers.

**Table 1 ijerph-19-08211-t001:** Comparison of baseline characteristics in T2D without metformin and with metformin cohorts.

Variables	T2D without Metformin		T2D with Metformin		SMD
	(*n* = 57,743)		(*n* = 57,743)		
	*n*	%	*n*	%	
Sex					
female	29,164	50.51	29,117	50.43	0.002
male	28,579	49.49	28,626	49.57	0.002
Age					
20–40	6633	11.49	6958	12.05	0.017
41–60	26,487	45.87	27,258	47.21	0.027
61–80	24,623	42.64	23,527	40.74	0.039
Obesity	1196	2.07	1242	2.15	0.006
Smoking status	1389	2.41	1435	2.49	0.005
Comorbidities					
Hypertension	31,946	55.32	33,796	58.53	0.065
Dyslipidemia	33,858	58.64	35,323	61.17	0.052
Coronary artery disease	16,168	28.00	16,191	28.04	0.001
Stroke	5879	10.18	5664	9.81	0.012
Atrial fibrillation	71	0.12	62	0.11	0.005
PAOD	2162	3.74	2095	3.63	0.006
CKD	3978	6.89	3530	6.11	0.031
Rheumatoid arthritis	1108	1.92	1085	1.88	0.003
Systemic lupus erythematous	157	0.27	124	0.21	0.012
Liver cirrhosis	1395	2.42	1424	2.47	0.003
Cancer	2899	5.02	2761	4.78	0.011
Psychosis	1219	2.11	1239	2.15	0.002
Depression	20,088	34.79	19,806	34.30	0.01
Dementia	1946	3.37	1775	3.07	0.017
COPD	11,241	19.47	11,176	19.35	0.003
Heart failure	2921	5.06	2917	5.05	0
Alcohol-related disorders	3219	5.57	3396	5.88	0.013
CCI					
1	11,989	20.76	11,214	19.42	0.034
2–3	28,937	50.11	30,508	52.83	0.054
>3	16,817	29.12	16,021	27.75	0.031
DCSI					
0	20,352	35.25	19,974	34.59	0.014
1	10,012	17.34	10,589	18.34	0.026
≥2	27,379	47.42	27,180	47.07	0.007
Medication					
SU	5919	10.25	5982	10.36	0.004
TZD	522	0.90	503	0.87	0.004
DPP-4 inhibitor	585	1.01	600	1.04	0.003
AGI	1366	2.37	1418	2.46	0.006
SGLT2i	16	0.03	4	0.01	0.016
Number of oral antidiabetic drugs					
1	56,805	98.38	56,861	98.47	0.039
2–3	924	1.60	865	1.50	0.039
>3	14	0.02	17	0.03	0.039
Insulin	21,207	36.73	21,239	36.78	0.001
Immunosuppressants	322	0.56	305	0.53	0.004
Statin	18,150	31.43	18,491	32.02	0.013
Aspirin	21,304	36.89	21,627	37.45	0.012
Influenza vaccination	12,121	20.99	12,369	21.42	0.011
Adult health examination	28,976	50.18	29,612	51.28	0.022
HbA1C > 2 times per year	88	0.15	91	0.16	0.001

SMD: standardized mean difference. A standardized mean difference of 0.1 or less indicates a negligible difference. T2D, type 2 diabetes; PAOD: Peripheral Arterial Occlusive Disease; CKD: Chronic Kidney Disease; COPD: Chronic Obstructive Pulmonary Disease; CCI, Charlson Comorbidity Index; DCSI, Diabetes Complication Severity Index; SU: Sulfonylureas; TZD: Thiazolidinedione; DPP-4: Dipeptidyl peptidase 4; AGI: Alpha-glucosidase inhibitors; SGLT2i: Sodium-glucose cotransporter 2 inhibitors; HbA1C: hemoglobin A1c.

**Table 2 ijerph-19-08211-t002:** Variables and the risk of asthma development.

Variables	New Development of Asthma						
	*n*	PY	IR	cHR	(95% CI)	aHR ^1^	(95% CI)
T2D without metformin	1440	235,580	6.11	1.00	(Reference)	1.00	(Reference)
T2D with metformin	2505	376,385	6.66	1.12	(1.05, 1.19) ***	1.13	(1.06, 1.2) ***
Sex							
female	2273	324,306	7.01	1.00	(Reference)	1.00	(Reference)
male	1672	287,660	5.81	0.82	(0.77, 0.88) ***	0.85	(0.79, 0.9) ***
Age							
20–40	347	83,608	4.15	1.00	(Reference)	1.00	(Reference)
41–60	1571	293,585	5.35	1.28	(1.14, 1.44) ***	1.27	(1.12, 1.43) ***
61–80	2027	234,772	8.63	2.05	(1.83, 2.3) ***	1.77	(1.55, 2.01) ***
Obesity	58	9665	6.00	0.9	(0.7, 1.17)	1.02	(0.79, 1.33)
Smoking	48	9542	5.03	0.75	(0.56, 0.99) *	0.88	(0.62, 1.24)
Comorbidities							
Hypertension	2363	328,017	7.20	1.28	(1.2, 1.36) ***	1.02	(0.95, 1.1)
Dyslipidemia	2180	348,665	6.25	0.92	(0.86, 0.98) **	0.9	(0.84, 0.97) **
Coronary artery disease	1418	165,550	8.57	1.51	(1.41, 1.61) ***	1.25	(1.15, 1.36) ***
Stroke	432	53,914	8.01	1.26	(1.14, 1.39) ***	1.05	(0.94, 1.18)
Atrial fibrillation	7	400	17.51	2.62	(1.25, 5.51) *	2.18	(1.03, 4.58) *
PAOD	158	19,645	8.04	1.24	(1.06, 1.45) **	1.06	(0.9, 1.25)
CKD	245	32,030	7.65	1.18	(1.04, 1.34) *	1.01	(0.88, 1.16)
RA	112	10,952	10.23	1.59	(1.32, 1.92) ***	1.37	(1.13, 1.66) **
SLE	5	1518	3.30	0.51	(0.21, 1.23)	0.45	(0.19, 1.09)
Liver cirrhosis	74	12,202	6.07	0.93	(0.74, 1.17)	0.88	(0.7, 1.12)
Cancer	155	22,538	6.88	1.05	(0.9, 1.24)	0.98	(0.82, 1.16)
Psychosis	81	11,822	6.85	1.06	(0.85, 1.32)	1.08	(0.86, 1.36)
Depression	1532	201,906	7.59	1.28	(1.2, 1.36) ***	1.09	(1.02, 1.18) *
Dementia	149	15,206	9.80	1.51	(1.28, 1.78) ***	1.1	(0.93, 1.31)
COPD	1280	110,472	11.59	2.17	(2.03, 2.31) ***	1.96	(1.82, 2.12) ***
Heart failure	327	26,773	12.21	1.95	(1.74, 2.19) ***	1.45	(1.28, 1.64) ***
Alcohol-related disorders	139	26,686	5.21	0.78	(0.66, 0.93) **	0.91	(0.74, 1.12)
CCI							
1	762	150,986	5.05	1	-		
2–3	1998	317,901	6.29	1.23	(1.13, 1.34) ***	0.95	(0.86, 1.04)
>3	1185	143,079	8.28	1.6	(1.46, 1.76) ***	0.88	(0.78, 0.99) *
DCSI							
0	1314	239,845	5.48	1	-		
1	716	114,542	6.25	1.13	(1.03, 1.24) **	0.96	(0.87, 1.05)
≥2	1915	257,579	7.44	1.34	(1.24, 1.43) ***	0.94	(0.85, 1.03)
Medications							
SU	527	71,531	7.37	1.18	(1.08, 1.29) ***	1.1	(1, 1.22) *
TZD	42	5412	7.76	1.2	(0.89, 1.63)	1.36	(0.95, 1.96)
DPP-4 inhibitor	10	2846	3.51	0.51	(0.27, 0.94) *	0.56	(0.29, 1.05)
AGI	80	12,851	6.23	0.95	(0.76, 1.19)	0.96	(0.75, 1.24)
Insulin	1567	211,239	7.42	1.24	(1.16, 1.32) ***	1.07	(1, 1.14)
Immunosuppressants	17	2407	7.06	1.07	(0.67, 1.72)	1.07	(0.66, 1.74)
Statin	996	162,800	6.12	0.91	(0.85, 0.98) *	0.8	(0.74, 0.87) ***
Aspirin	1553	208,758	7.44	1.24	(1.16, 1.32) ***	0.96	(0.89, 1.04)

PY: person-years; IR: incidence rate per 1000 person-years; cHR: crude hazard ratio; aHR: adjusted hazard ratio. ^1^: adjusted by sex, age, comorbidities, and medication. *: *p*-value < 0.05; ** *p* < 0.01, *** *p* < 0.001. PAOD: Peripheral Arterial Occlusive Disease; CKD: Chronic Kidney Disease; RA: Rheumatoid arthritis; SLE: Systemic lupus erythematosus; COPD: Chronic Obstructive Pulmonary Disease; SU: Sulfonylureas; TZD: Thiazolidinedione; DPP-4: Dipeptidyl peptidase 4; AGI: Alpha-glucosidase inhibitors.

**Table 3 ijerph-19-08211-t003:** Hazard ratios (HRs), and 95% confidence intervals (CIs) for outcome disease among the sampled patients.

Outcome	T2DM without Metformin			T2DM with Metformin								
	*n*	PY	IR	*n*	PY	IR	cHR	(95% CI)	*p*-Value	aHR ^1^	(95% CI)	*p*-Value
New development of asthma	1440	235,580	6.11	2505	376,385	6.66	1.12	(1.05, 1.19)	<0.001	1.13	(1.06, 1.2)	<0.001
Acute exacerbation of asthma	163	241,618	0.67	393	389,209	1.01	1.59	(1.32, 1.91)	<0.001	1.62	(1.35, 1.95)	<0.001
Hospitalization for asthma	124	241,947	0.51	328	390,263	0.84	1.52	(1.23, 1.87)	<0.001	1.5	(1.22, 1.85)	<0.001

PY: person-years; IR: incidence rate per 1000 person-years; cHR: crude hazard ratio; aHR: adjusted hazard ratio. ^1^: adjusted by sex, age, comorbidities, and medications.

**Table 4 ijerph-19-08211-t004:** Hazard ratios of outcomes associated with metformin use.

**Variables**	**New Development of Asthma**						
	** *n* **	**PY**	**IR**	**cHR**	**(95% CI)**	**aHR ^1^**	**(95% CI)**
Non-use of metformin	1440	235,580	6.11	1.00	(Reference)	1.00	(Reference)
Metformin of drug days							
28–364	593	91,935	6.45	1.05	(0.95, 1.15)	1.03	(0.93, 1.13)
364–728	364	65,236	5.58	0.91	(0.81, 1.02)	0.93	(0.83, 1.05)
>728	1548	219,214	7.06	1.23	(1.14, 1.32) ***	1.24	(1.16, 1.34) ***
**Variables**	**Acute Exacerbation of Asthma**						
	** *n* **	**PY**	**IR**	**cHR**	**(95% CI)**	**aHR ^1^**	**(95% CI)**
Non-use of metformin	163	241,618	0.67	1.00	(Reference)	1.00	(Reference)
Metformin of drug days							
28–364	97	94,395	1.03	1.51	(1.17, 1.94) **	1.56	(1.21, 2.01) ***
364–728	53	66,695	0.79	1.16	(0.85, 1.58)	1.26	(0.93, 1.73)
>728	243	228,118	1.07	1.78	(1.46, 2.18) ***	1.77	(1.44, 2.17) ***
**Variables**	**Hospitalization for Asthma**						
	** *n* **	**PY**	**IR**	**cHR**	**(95% CI)**	**aHR ^1^**	**(95% CI)**
Non-use of metformin	124	241,947	0.51	1.00	(Reference)	1.00	(Reference)
Metformin of drug days							
28–364	76	94,594	0.80	1.59	(1.19, 2.12) **	1.59	(1.2, 2.12) **
364–728	43	66,792	0.64	1.28	(0.91, 1.82)	1.36	(0.96, 1.92)
>728	209	228,877	0.91	1.55	(1.24, 1.95) ***	1.5	(1.19, 1.89) ***

PY: person-years; IR: incidence rate per 1000 person-years; cHR: crude hazard ratio; aHR: adjusted hazard ratio. ^1^: adjusted by sex, age, comorbidities, and medications. ** *p* < 0.01, *** *p* < 0.001.

## Data Availability

Data of this study are available from the National Health Insurance Research Database (NHIRD) published by Taiwan National Health Insurance (NHI) Administration. The data utilized in this study cannot be made available in the paper, the Appendix A, or in a public repository due to the “Personal Information Protection Act” executed by Taiwan government starting from 2012. Requests for data can be sent as a formal proposal to the NHIRD Office (https://dep.mohw.gov.tw/DOS/cp-2516-3591-113.html (accessed on 8 December 2021)) or by email to stsung@mohw.gov.tw.

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
