# Peer review of "Metformin and the Development of Asthma in Patients with Type 2 Diabetes"

_ijerph, 2022, doi:10.3390/ijerph19138211_

Round 1
Reviewer 1 Report
This is an interesting work analyzing clinical information from a large cohort of Taiwanese T2D patients. The dataset is richer than previously published works. The authors compare the risk of asthma development and exacerbation between metformin users and non-users in Taiwan (Chinese population) from data collected over ~17 years. The results markedly contradict previous studies (that are mentioned in the manuscript). The authors find that metformin is associated to an increased risk for asthma onset, exacerbation, and hospitalization with Cox PH models.
I would like to ask the authors to please address the below comments:
1. In Table 1, below the Age factor(s), there is a line “Mean, (SD)” which I assume it should not have been there. Please correct or explain what this factor represents.
2. Table 1 contains the balanced “T2D with Metformin” vs “T2D without Metformin” design (57,743 patients in each group). These patients are the propensity-matched subsample and as explained at lines 126-127 (p. 4), “The nearest-neighbor algorithm was used to match pairs, assuming the standardized mean difference (SMD) < 0.1 to be a negligible difference between the study and 127 control cohorts.”. However, from the table we see that the SMD of several factors exceeds this threshold. My interpretation is that the matching did not work as expected:
A) Please show details on the formula used for the propensity matching for the reader to understand the methodology and, if exist, its pitfalls.
B) Please provide evidence on how well the matching worked. For example, are there significant differences in the frequencies of Sex and Age (e.g. are women tend to be older in you cohort? Are men of women or some age interval is more associated to a certain comorbidity?). Please check in more details such comparisons to evaluate the outcome of the matching algorithm.
3. The study focusses on metformin use / not use but we do not have any evidence on the doses and how well the patients followed the prescription etc. Is it possible that there is a hidden factor there (eg doses over time or doses across different groups) affecting the validity of the results?
4. Are there any studies that agree with the findings reported here? Despite the methodological problems of the previous studies (sample size, matching, primary vs secondary outcome) and the differences in populations (UK vs Chinese), all of them mentioned here show that metformin is associated with a lower asthma risk (onset and exacerbation). Although it is exciting, I find this suspicious, and I would like to see more evidence at:
A) How well the matching worked (see above)
B) Please show that your results are reproducible in smaller sets of your data. Based on the results of Table 2, I would suggest appropriately matched men only, women only and each of the three different age groups after adjusting for the other factors (as it was done in the manuscript).
Author Response
Thank you for your reveiwng of our manuscript, and your recommdedations make our paper more complte. We reply your comments with the attached response letter.

Reviewer 2 Report
The MS addresses an important issue of metformin off-target effects in the Taiwanese population and present contradictory results (as compared to other published data) on metformin-treated T2DM subjects by showing an increase of asthma incidence. This challenges the prevailing view of the beneficial effects of metformin on chronic inflammation in general incl. asthma.
I have two major concerns that need to be addressed:
1. Metformin represented (and still is in fact despite the promising results on SGLT-2 inhibitors) the first-line drug at the time of enrollment/study period. What was the reason not to prescribe metformin to newly diagnosed T2DM when subjects were matched for T2DM "severity"? The group of metformin non-users has to be carefully described in order to avoid event. bias.
2. Is incident asthma in all cases atopic? What information on pathogenesis authors have? Any seasonal effects detected?
Author Response
Responses to the comments of Reviewer #2
The MS addresses an important issue of metformin off-target effects in the Taiwanese population and present contradictory results (as compared to other published data) on metformin-treated T2DM subjects by showing an increase of asthma incidence. This challenges the prevailing view of the beneficial effects of metformin on chronic inflammation in general incl. asthma.
I have two major concerns that need to be addressed:
- Metformin represented (and still is in fact despite the promising results on SGLT-2 inhibitors) the first-line drug at the time of enrollment/study period. What was the reason not to prescribe metformin to newly diagnosed T2DM when subjects were matched for T2DM "severity"? The group of metformin non-users has to be carefully described in order to avoid event. bias.
Response: Thank you for your encouragement and recommendations. Most of our patients were stopped metformin due to abdominal discomfort. But the Taiwan’s 2019 Diabetes Atlas (Chu CH, et al. Trends in antidiabetic medical treatment from 2005 to 2014 in Taiwan. J Formos Med Assoc. 2019;118 Suppl 2:S74-S82.)) revealed that, although metformin was the recommended first-line treatment for T2D, there were more patients using sulfonylureas than metformin from 2005 to 2008 (attached table). Because our study was conducted from 2000 to 2018, we believe that many physicians still prefer to use sulfonylurea to treat T2D in 2000 to 2008 for its fast hypoglycemic effect. We have included this information on page 10 to make readers to realize the group of metformin nonusers in this study.
- Is incident asthma in all cases atopic? What information on pathogenesis authors have? Any seasonal effects detected?
Response: We agree with your opinions that the phenotype and endotype of asthma are complex and represent a multitude of pathogenesis, including eosinophilic and non-eosinophilic asthma, non-allergic eosinophilic inflammation, mixed granulocytic asthma, type 1 and type 17 neutrophilic inflammation (Papi A et al., Asthma. Lancet. 2018 24;391(10122):783-800). However, as our database does not have access to different immune cells from the sputum or blood, we cannot distinguish the various types of asthma (page 11). Thank you for your suggestion to make our manuscript more complete.

Round 2
Reviewer 1 Report
Dear authors,
Thank you for addressing my comments. The study is well-described and several parts of the analysis have been clarified.
One final suggestion is to make the Discussion section shorter by moving some paragraphs to the literature review part (eg those referring to metformin functions, not exclusively related to asthma). The comparison to other studies, the advantages and disadvantages of your study and others should be the highlights of that section.
I have no further comments.
Author Response
Dear reviewer: Thank you for your recommendations. We have moved the paragraph (not related to asthma) to the literature review part, to make the Discussion section shorter.
Reviewer 2 Report
The authors addressed the criticism adequately. I have no further comments.
Author Response
Thank you for reviewing of our manuscript.